# Incidence and risk factors of calcium channel blocker–related edema in hypertensive patients: A Multicenter retrospective cohort study

Koricho Simie Tolla[1], Gashaw Solela[2*], Getachew Wondafrash[3], Abay Burusie[4], Gebi Agero[4], Dureti Desta Garoma[4], Wubshet Abraham Alemu[1], Bereket Sinshaw Engida[5], Surafel Mekasha Woldeyes[4], Berhanu Moges Abera[2], Mulualem Alemayehu Gebreselassie[6]

1 Department of Internal Medicine, College of Health Sciences, Arsi University, Asella, Ethiopia, 2 Department of Cardiology, College of Health Sciences, Addis Ababa University, Addis Ababa, Ethiopia, 3 Division of Nephrology, Department of Internal Medicine, Yekatit 12 Hospital Medical College, Addis Ababa, Ethiopia, 4 Department of Public Health, College of Health Sciences, Arsi University, Asella, Ethiopia 5 Yehuleshet Specialty Clinic, Addis Ababa, Ethiopia, 6 ICMC General Hospital, Addis Ababa, Ethiopia

* gashawsol@gmail.com

## Abstract

### Background

Hypertension is a major risk factor for cardiovascular disease and remains the leading cause of mortality worldwide. Calcium channel blockers (CCBs) are commonly used to lower blood pressure because they are effective and affordable. However, CCBs can cause vasodilatory adverse effects, including peripheral edema, which may lead to additional therapy and affect adherence. This study assessed the incidence and risk factors of CCB-related edema among hypertensive patients in Ethiopia.

### Methods

This retrospective multicenter cohort study involved interviews and reviews of medical records of adults (aged ≥18 years) with essential hypertension who were prescribed calcium channel blockers (CCBs) between July 15 and August 14, 2025. A total of 292 participants were selected using systematic random sampling. A structured questionnaire was used to collect sociodemographic and clinical data. Descriptive statistics summarized baseline characteristics. Time-to-event analysis with the log-rank test assessed the duration from CCB initiation to edema onset. Binary and multivariate logistic regression analyses identified factors associated with CCB-related edema, and a $p$-value <0.05 was considered statistically significant.

**Data availability statement:** All relevant data are within the manuscript and its Supporting information files.

**Funding:** The author(s) received no specific funding for this work.

**Competing interests:** The authors have declared that no competing interests exist.

## Results

Among 292 participants (mean age 58.2 years; 53.4% female), 20.9% had diabetes mellitus and 16.8% had dyslipidemia. Amlodipine was the most frequently prescribed CCB (94.8%). Peripheral edema developed in 38.7% of patients, with a mean onset time of 8.3 weeks. In multivariate logistic regression analysis, only longer daily standing duration was significantly associated with edema (AOR = 1.92; 95% confidence interval: 1.03–3.58; $p = 0.041$). Time-to-event analysis showed a progressive increase in edema risk with continued CCB use. Patients receiving amlodipine 10 mg daily had a greater (42.5% vs. 33%) and earlier risk of edema than those on 5 mg amlodipine daily (log-rank $p = 0.003$).

## Conclusions

Calcium channel blocker–related peripheral edema is common among Ethiopian patients with hypertension and is more likely with higher doses and prolonged daily standing. Clinicians should be aware of its high prevalence to provide effective patient counseling and avoid unnecessary investigations or treatments, such as diuretics.

## Introduction

Hypertension is defined as a persistent systolic blood pressure (SBP) of at least 140 mmHg or a diastolic blood pressure (DBP) of at least 90 mmHg [1]. If left untreated, it is associated with an increased risk of cardiovascular disease (CVD) events, including coronary heart disease, heart failure, stroke, and death [2]. The global prevalence of hypertension has doubled over the past three decades, with nearly two-thirds of affected individuals residing in low- and middle-income countries (LMICs) [1,3]. Despite this high burden, treatment and control rates remain low in many regions, particularly in sub-Saharan Africa [3]. In Ethiopia, hypertension has become a growing public health problem, with urban prevalence among adults reported to be as high as 30.3% [4].

Evidence from randomized controlled trials (RCTs) has consistently shown that antihypertensive therapy reduces cardiovascular morbidity and mortality [5]. The United States Eighth Joint National Committee (JNC 8) guidelines recommend that initial therapy should include a thiazide diuretic, calcium channel blocker (CCB), angiotensin-converting enzyme (ACE) inhibitor, or angiotensin receptor blocker (ARB) [6]. Among these, CCBs and ARBs are the most frequently prescribed drug classes worldwide. Multiple studies have identified CCBs as the most commonly used antihypertensive medications [7]. Similarly, studies from Ethiopia have identified CCBs, along with ACE inhibitors and thiazide diuretics, as the most commonly prescribed antihypertensive agents [8,9].

Dihydropyridine (DHP) CCBs are associated with vasodilatory adverse effects, including peripheral edema [10]. This edema is thought to result from increased

capillary hydrostatic pressure due to greater dilation of pre-capillary arterioles compared to post-capillary venules [11]. Peripheral edema not only causes discomfort but may also lead to additional investigations, the prescription of loop diuretics, and poor adherence to therapy. Patients on DHP-CCBs are reported to be up to 60% more likely to receive loop diuretics than those on other antihypertensives [10,12].

The reported incidence of CCB-induced peripheral edema varies widely. Meta-analyses of RCTs have shown a pooled incidence of around 10.7% [13], while some observational studies have reported rates as high as 29% [14,15]. This variation may reflect differences in study design, populations, and environmental or genetic factors. Several potential risk modifiers have been identified, including CCB type, dosage, duration of use, age, sex, comorbidities (such as diabetes, dyslipidemia, chronic obstructive pulmonary disease, cerebrovascular disease, and coronary artery disease), and upright posture [13,16–18]. Exposure to environmental heat has also been mentioned to be associated with the development of edema [19], which can possibly explain the variation in reported incidence from different geographic locations. However, the influence of many of these factors remains inconsistent across studies.

Given the lack of local data and potential differences in patient characteristics, this study aimed to determine the incidence and identify risk factors of CCB–related peripheral edema among hypertensive patients treated at two teaching hospitals in Ethiopia. We expect the findings from this study to raise awareness among clinicians, improve patient counseling, and help reduce unnecessary investigations and additional drug therapy related to CCB-induced edema.

## Materials and methods

### Ethics approval and consent to participate

Ethical clearance was obtained from the College of Health Science, Arsi University, Ethical Review Committee (25/03/2025, protocol no. A/CHS/RC/152/2025). Written informed consent was obtained from all participants. Data were anonymized to protect confidentiality, and only relevant research information was collected. Clinical care and research activities were strictly separated, with treating physicians not involved in data collection or analysis. No therapeutic interventions were performed. As this was a non-interventional study, a clinical trial number was not required. The study adhered to the Declaration of Helsinki.

### Study design, area, and period

We conducted a multicenter retrospective cohort study by reviewing medical records and interviewing adult patients (aged ≥18 years) diagnosed with essential hypertension who were prescribed CCBs. The study was carried out from July 15 to August 14, 2025, at Asella Teaching and Referral Hospital (ATRH) and Yekatit 12 Hospital Medical College (Y12HMC). We selected these two hospitals to capture diverse patient populations. ATRH serves suburban and rural communities, while Y12HMC serves an urban population. Both institutions have well-established hypertension follow-up clinics that provide continuous care, maintain comprehensive medical records, and are staffed by residents, internists, fellows, and subspecialists, ensuring reliable sources of clinical and patient information for this study.

### Source and study population

The source population included all adult patients (aged ≥18 years) diagnosed with essential hypertension and attending follow-up clinics at ATRH and Y12HMC who had been prescribed CCBs as part of their treatment regimen. The study population comprised those patients who attended follow-up during the study period and had complete medical records. We excluded patients who were non-compliant with treatment or if data on CCB usage or peripheral edema were missing.

### Eligibility criteria

Adult patients aged 18 years and above, diagnosed with essential hypertension and prescribed CCBs for at least six months, or who discontinued CCB therapy earlier due to the development of peripheral edema, were eligible for inclusion.

All types of CCBs, including amlodipine and nifedipine, were considered. We excluded patients with pre-existing conditions that could contribute to edema, such as heart failure, renal insufficiency, deep vein thrombosis, venous insufficiency, chronic liver disease, hypothyroidism, pregnancy, or other relevant conditions. Additionally, patients with incomplete medical records regarding history, CCB use, or development of edema, as well as those unable to provide informed consent, were not included in the study.

**Sample size determination and sampling technique**

The sample size was calculated based on a Nigerian prospective cohort study reporting a 29% incidence of peripheral edema among patients prescribed amlodipine or nifedipine [15]. Using a single population proportion formula with 95% confidence and 80% power, the initial sample size was 316. Considering that high-dose CCBs increase the risk of edema 2.8-fold [13], Fleiss' formula with continuity correction yielded 318. With a total hypertensive patient population of approximately 2,500 at both centers, and 50% on CCB therapy, the source population was estimated at 1,250. Applying the finite population correction resulted in an adjusted sample size of 253. Accounting for a 10% non-response rate, the final sample size was 292.

During the actual study, out of 2570 patients with hypertension who visited referral clinics, 1,336 were identified as having ever used CCBs. Of these, 1,234 patients who had never used CCBs were excluded, along with an additional 31 patients who had used CCBs but had incomplete data. Among the remaining 1,305 patients, 54 were excluded because they had used CCBs for less than six months without developing edema. Ultimately, the final source population consisted of 1,251 hypertensive patients who had used CCBs, had complete data, and either adhered to CCB therapy for at least six months or developed CCB-related edema at some point (see flow chart in Fig 1).

Of the total sample, 52% (n = 152) were recruited from ATRH and 48% (n = 140) from Y12HMC. Systematic random sampling was used to select patients attending referral clinics during the study period. If the sampled patient is not a CCB user, the CCB user following him/her was selected as a sample.

**Study variables**

The dependent variable was the incidence of calcium channel blocker (CCB)-related peripheral edema. The independent variables included sociodemographic factors (age, gender, and occupation), duration of daily upright posture, comorbid conditions (diabetes, chronic obstructive pulmonary disease, cerebrovascular accidents, coronary artery disease, dyslipidemia, and others), body mass index, and medication history. Medication history encompassed the type and dosage of CCBs prescribed, duration of therapy before discontinuation or edema development, and the use of other antihypertensive medications, including ACE inhibitors, angiotensin receptor blockers, thiazides, and beta-blockers. Among these, the specific dosage of CCB was considered the primary independent variable.

**Operational definitions**

**CCB user.** An adult patient with essential hypertension who has used a CCB for at least six months or who discontinued CCB therapy due to the development of peripheral edema. The primary CCBs in this study were amlodipine and nifedipine.

**High-dose CCB therapy.** The use of amlodipine >5 mg/day or nifedipine ≥40 mg/day.

**Posture.** The predominant body position maintained during daily activities, classified as upright, sitting, or recumbent/lying down.

**Adherence to salt restriction.** Consumption of ≤5 g of table salt per day (approximately one teaspoon), in line with World Health Organization recommendations.

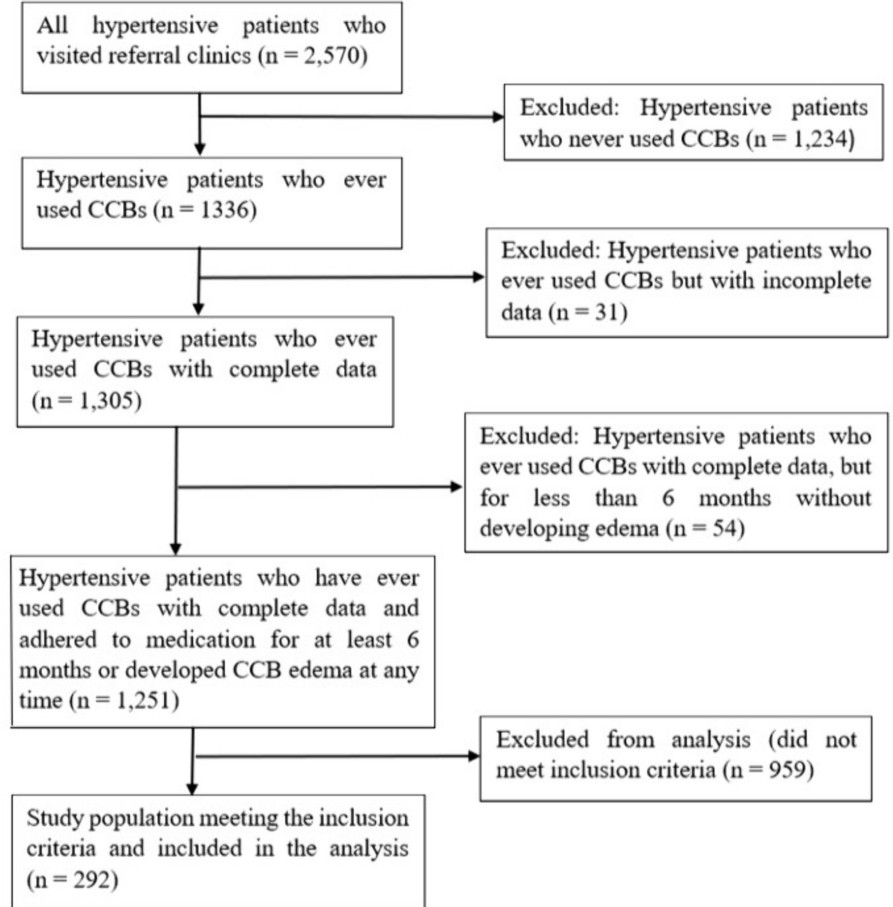

**Fig 1. Study flowchart of hypertensive patient selection and the study population for assessing calcium channel blocker–related peripheral edema at ATRH and Y12HMC, Ethiopia, 2025.**

## Data collection procedure

Data were collected using a structured questionnaire administered through the Kobo Toolbox platform. Demographic information and selected details on edema detection and progression were obtained via patient interviews, while comorbidities, physical examination findings, and medication histories were extracted from medical records. Data collection was performed by a team of four internal medicine residents and two internists, under the supervision of an internist, a cardiology fellow, and a nephrologist. Despite differences in training levels, the assessment of edema and completion of the questionnaire were considered straightforward, minimizing potential inter-observer variability.

## Data analysis

Data were checked for completeness and analyzed using SPSS version 27 (IBM Corporation, Armonk, NY, USA). Descriptive statistics, including frequency distributions and measures of central tendency and dispersion, were used to summarize demographic and clinical characteristics of the study population. Time-to-event (survival) analysis was performed to evaluate the interval from initiation of CCB therapy to the development of peripheral edema, stratified by CCB dose, with patients who did not develop edema during follow-up being censored. Binary logistic regression was used

to estimate crude odds ratios (COR) for the occurrence of edema. Multivariable logistic regression was then performed to adjust for potential confounders and assess the independent associations of multiple risk factors with the outcome, reported as adjusted odds ratios (AOR). Statistical significance was defined as a p-value <0.05.

### Data quality assurance

A pretest was conducted on 5% (n = 15) of the medical records to evaluate the clarity, feasibility, and consistency of the questionnaire. Data were collected by trained physicians under the supervision of the principal investigators. The completeness, consistency, and accuracy of the data were regularly monitored, and any discrepancies were promptly addressed to minimize information bias and ensure data reliability.

## Results

### Sociodemographic data

A total of 292 patients met the inclusion criteria and were included in the analysis (Fig 1). The mean age was 58.2 years, and 53.4% were female. Table 1 shows demographic characteristics of participants. The mean age was similar between participants who developed peripheral edema and those who did not (58.27 vs. 58.22 years). The mean duration of hypertension was 47.9 months, ranging from 1 to 360 months. Regarding occupation, 37% were retired, 20.5% were farmers, 16.6% were teachers, 14% were unemployed, and 10.3% were merchants. On average, participants reported spending 2.9 hours per day standing (standard deviation [SD] = 2.0; range = 0–10 hours). The mean body weight was 68.0 kg (SD = 11.8; range = 40–115 kg).

**Table 1. Socio-demographic characteristics and comorbidities of hypertensive patients taking CCB at ATRH and Y12HMC, Ethiopia, 2025 (N = 292).**

| Variable | Number | Percentage |
|---|---|---|
| Total patients (n) | 292 | 100.0 |
| Sex | | |
| Female | 156 | 53.4 |
| Male | 136 | 46.6 |
| Occupation | | |
| Retired | 108 | 37.0 |
| Farmer | 60 | 20.5 |
| Teacher | 48 | 16.6 |
| Merchant/Trader | 30 | 10.3 |
| Office worker | 21 | 7.2 |
| Others | 25 | 8.6 |
| Comorbidities | | |
| Diabetes mellitus | 61 | 20.9 |
| Dyslipidemia | 49 | 16.8 |
| Stroke | 28 | 9.6 |
| Ischemic heart disease | 25 | 8.6 |
| Neuropathy | 24 | 8.2 |
| HIV Infection | 9 | 3.1 |
| Chronic kidney disease | 2 | 0.7 |
| Psychiatric condition | 2 | 0.7 |

Comorbidities were common, with diabetes mellitus in 20.9%, dyslipidemia in 16.8%, stroke in 9.6%, ischemic heart disease in 8.6%, and neuropathy in 8.2% of patients. Chronic kidney disease, HIV infection, and other conditions were less frequent, while 49.7% of participants had no additional medical conditions (Table 1). At the time of the most recent visit or edema assessment, blood pressure was controlled in 64% of participants, with a mean systolic BP of 134.6 mmHg (range 80–207) and a mean diastolic BP of 78.9 mmHg (range 51–130).

## Medication history and adverse events

The most commonly prescribed CCB was amlodipine (95%), followed by nifedipine (4.5%). High-dose CCB was administered to 57.2% of patients (Fig 2). Concomitant antihypertensive therapy was used in 57.2% of participants, with ACE inhibitors prescribed to 49.1% and thiazide diuretics to 26.9% of these patients (Table 2).

No patients developed edema within the first week of CCB therapy. During follow-up, 38.7% of patients developed peripheral edema, with mean of 10.5 weeks (SD: 8.3 weeks) after initiating treatment. Edema was predominantly bilateral and occurred more often in patients on high-dose therapy compared to low-dose therapy (42.5% vs. 33%). Patients self-reported edema in 84.9% of cases. Other commonly reported adverse effects included headache (14%), constipation (11.3%), and dizziness or lightheadedness (9.6%).

The mean change of SBP, from baseline to the development of edema or to the last evaluation in those without edema, was 29.86 mmHg (SD = 19.02), ranging from −20–112 mmHg. The mean change in those who did not develop CCB-related edema was 30.35 mmHg (SD = 19.69), while in those who did was 29.08 mmHg (SD = 17.97).

## Risk factors

Binary logistic regression showed that longer daily standing duration and high-dose CCB therapy were associated with edema development. In multivariate analysis, only daily standing duration remained significant (AOR: 1.92; 95% CI: 1.03–3.58), while high-dose CCB approached significance (AOR: 1.58; 95% CI: 0.97–2.57). No comorbid conditions or other adverse effects, including headache and constipation, were associated with edema (Table 3). The mean change of

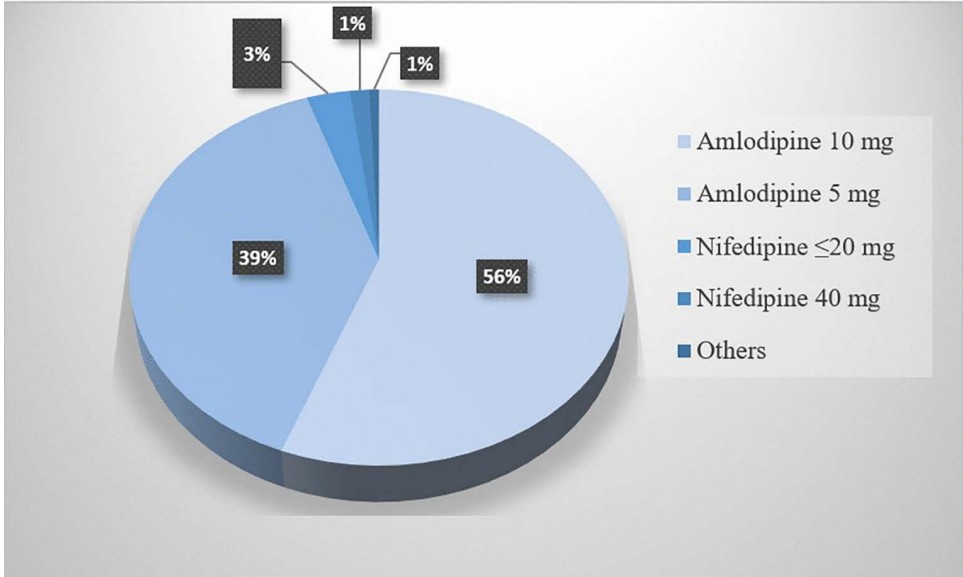

**Fig 2. Pie chart for the types and daily doses of CCBs used among hypertensive patients at ATRH and Y12HMC, Ethiopia, 2025 (N = 292).**

**Table 2. Additional Antihypertensives used among hypertensive patients taking CCB at ATRH and Y12HMC, Ethiopia, 2025 (N = 292).**

| Concomitant antihypertensive drugs | Frequency | Percentage |
|---|---|---|
| ACEi | 82 | 49.1 |
| Thiazide diuetics | 45 | 26.9 |
| ACEi + Thiazide diuretics | 20 | 12.0 |
| ARBs | 6 | 3.6 |
| ARBs + Thiazide diuretics | 5 | 3.0 |
| ACEi + BB | 3 | 1.8 |
| ACEi + BB + others | 2 | 1.2 |
| Other combinations | 4 | 2.4 |
| Total | 167 | 100.0 |

**Abbreviations:** ACEi, angiotensin-converting enzyme inhibitor; ARB, angiotensin receptor blocker; BB, beta-blocker.

**Table 3. Logistic regression on risk factors for edema among hypertensive patients taking CCB at ATRH and Y12HMC, Ethiopia, 2025 (N = 292).**

| Variable | COR (95% CI) | p value | AOR (95% CI) | p value |
|---|---|---|---|---|
| Age | 1.01 (0.99 - 1.03) | 0.480 | – | – |
| Sex(male) | 1.60 (0.92 - 2.80) | 0.099 | – | – |
| Duration of daily Standing | 1.2 (1.02 - 1.30) | 0.020 | 1.92 (1.03 - 3.58) | 0.041 |
| High-dose CCB | 1.45 (1.018 - 2.06) | 0.039 | 1.58 (0.97 - 2.57) | 0.068 |
| Use of other antihypertensive medications | 0.60 (0.36 - 1.02) | 0.060 | – | – |
| Controlled BP | 0.72 (0.44 - 1.17) | 0.180 | – | – |
| Systolic blood pressure change from baseline | 0.99 (0.98 - 1.01) | 0.580 | – | – |
| Other CCB side effects | 1.93 (0.67 - 5.56) | 0.230 | – | – |
| Physical exercise | 1.66 (0.55 - 4.99) | 0.370 | – | – |
| Adherence to salt restriction | 1.37 (1.56 - 3.35) | 0.490 | – | – |
| Current smoking status | 0.67 (.062 - 7.18) | 0.740 | – | – |
| Daily Alcohol consumption | 4.91 (0.74 - 32.66) | 0.100 | – | – |

**Abbreviations:** AOR, Adjusted Odds Ratio; BP, blood Pressure; CCB, calcium channel blocker; COR, Crude Odds Ratio.

SBP was similar for both groups and the magnitude of change in SBP after initiation of CCB was not associated with the development of edema (see S1 File).

### Management of CCB-related edema

Following edema onset, management strategies varied. Drug adjustment, including discontinuation, was performed in 58.4% of patients, and additional investigations were conducted in 23.9%. Diuretics were added in 22.3% of cases, whereas no patients received ACE inhibitors or ARBs. A minority of patients (17.7%) continued therapy with conservative measures, such as lifestyle modifications.

### Time-to-event analysis

Time-to-event analysis, stratified by daily CCB dose (high versus low), showed a progressive increase in the risk of edema with longer duration of CCB use. Participants receiving amlodipine 10 mg experienced a higher and earlier risk of edema

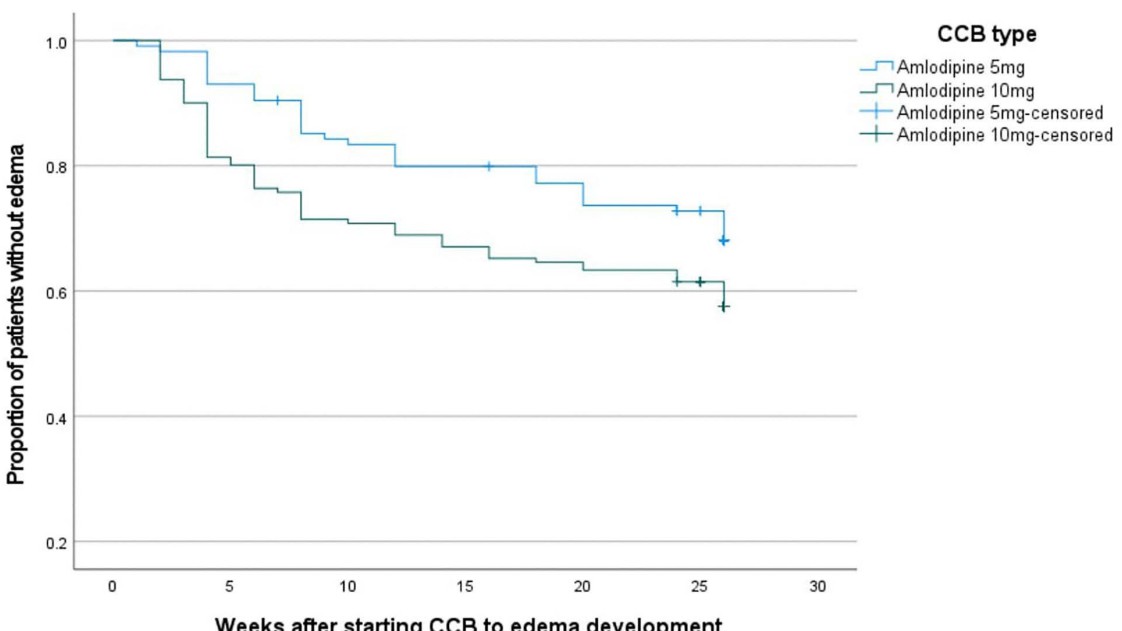

**Fig 3. Kaplan-Meier curve for development of edema among hypertensive patients taking CCB at ATRH and Y12HMC, Ethiopia, 2025 (N = 292).**

compared to those taking 5 mg (Fig 3). The difference between the groups was statistically significant by the log-rank test (P = 0.003; Log Rank [Mantel-Cox] = 15.703). Analysis was restricted to participants taking amlodipine, which accounted for nearly 95% of the study population (see S2 File).

## Discussion

We retrospectively examined the incidence of CCB-related edema among 292 patients with hypertension at two hospitals in Ethiopia. The mean age of 58.2 years was comparable to that reported in one of the large meta-analyses by Makani et al [13]. We found the incidence of CCB-related edema amongst Ethiopians is higher (38.7%) than the range of 10.7% to 29% reported from elsewhere, and in meta-analyses [13–15].

The average time to identify edema in our study was 10.5 weeks (73.5 days), which is comparable to the Indian study reporting 69 days [14]. A different study reported a much shorter duration, 27.3 days [20]. This difference may be explained by the fact that 85% of cases of CCB- related edema were reports by patients, which may result in delayed recognition. This is supported by the fact that edema was identified as early as 1 week after initiating CCB in this study. This suggests that active monitoring for edema by physicians after initiating CCBs could lead to earlier detection.

In agreement with several other studies, this study also showed significantly higher incidence of edema in patients using higher doses of CCB, particularly in participants taking amlodipine 10 mg compared to 5 mg [13]. There is also data supporting a progressive increase in edema incidence with longer duration of CCB use, as observed in our study, as illustrated by the Kaplan-Meier curve in the results section [15]. Not just higher incidence, this curve also showed earlier onset of edema among high-dose CCB users. This is supported by the biological plausibility of high-dose CCB resulting in a higher effect on precapillary vessels and subsequently increasing capillary hydrostatic pressure and edema.

Several modifiers of CCB-related edema have been well reported by studies, including upright posture, warmth, older age, and female gender [21]. Our Study showed a significant association with the duration of upright posture. This is due to the upright posture providing an additional gravitational contribution to increased hydrostatic pressure and the

development of edema [16,17]. One study showed an association of CCB edema with comorbid conditions, including diabetes, COPD, coronary artery disease, and dyslipidemia, which our study didn't evidence. This is likely due to the fact that, except for diabetes, other comorbid conditions were less frequent in our study.

The concomitant use of renin–angiotensin system inhibitors, particularly ACE inhibitors, has been shown to reduce both the incidence and severity of calcium channel blocker–related edema [13,16,17]. In our study, coadministration of an ACEi and/or a thiazide diuretic was associated with a trend toward a lower risk of CCB-induced edema (odds ratio 0.60; 95% CI, 0.36–1.02); however, this association did not reach statistical significance, likely reflecting the limited number of participants receiving these agents.

Lack of acknowledgment of edema as an adverse effect of CCBs by clinicians might result in inadvertent use of additional interventions. One of the adverse effects of CCB-related edema is the sequential use of diuretics, which may expose patients to drug-related side effects like acute kidney injury. A large cohort study reported that CCB users experienced 60% higher rates of being subsequently dispensed a loop diuretic compared to other antihypertensive drug users [12]. Although we cannot directly compare this figure, our study found that 22.3% of participants were prescribed to use diuretics. Additionally, nearly 24% of our patients underwent additional laboratory and imaging tests after the detection of edema.

Our study has some limitations. First, as a retrospective cohort study, it relied on existing medical records, which may have limited the inclusion to patients with complete data and could potentially underestimate the true incidence of CCB-related edema. The diagnosis of edema was based on routine clinical judgment without standardized criteria, which may have introduced some variability in assessment. Finally, because there was no control group, we cannot directly compare the incidence of edema between patients on CCBs and those on other antihypertensive medications.

## Conclusion

CCB–related peripheral edema is common among Ethiopian patients with hypertension and is more likely with higher doses and prolonged daily standing. Clinicians should be aware of its high prevalence to provide effective patient counseling and avoid unnecessary investigations or treatments, such as diuretics.

## Supporting information

**S1 File. Complete raw dataset in SPSS format.**
(SAV)

**S2 File. Processed dataset derived from the full dataset, including recategorized variables, computed measures, and variables prepared for time-to-event analysis of edema development.**
(SAV)

## Acknowledgments

The authors extend their gratitude to the study participants and the healthcare personnel at ATRH and Y12HMC referral clinics for their valuable support.

## Author contributions

**Conceptualization:** Koricho Simie Tolla, Abay Burusie, Gebi Agero.

**Formal analysis:** Gashaw Solela, Getachew Wondafrash, Abay Burusie, Gebi Agero, Dureti Desta Garoma, Wubshet Abraham Alemu, Bereket Sinshaw Engida, Surafel Mekasha Woldeyes, Berhanu Moges Abera, Mulualem Alemayehu Gebreselassiea.

**Investigation:** Koricho Simie Tolla.

**Methodology:** Koricho Simie Tolla, Gashaw Solela, Getachew Wondafrash, Abay Burusie, Gebi Agero.

**Supervision:** Koricho Simie Tolla, Gashaw Solela, Getachew Wondafrash.

**Validation:** Dureti Desta Garoma, Wubshet Abraham Alemu, Bereket Sinshaw Engida, Surafel Mekasha Woldeyes, Berhanu Moges Abera, Mulualem Alemayehu Gebreselassiea.

**Writing – original draft:** Koricho Simie Tolla.

**Writing – review & editing:** Gashaw Solela, Getachew Wondafrash, Abay Burusie, Gebi Agero, Dureti Desta Garoma, Wubshet Abraham Alemu, Bereket Sinshaw Engida, Surafel Mekasha Woldeyes, Berhanu Moges Abera, Mulualem Alemayehu Gebreselassiea.

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
