## [Decision Letter · Decision Letter 0]

1 Dec 2025

PONE-D-25-58381Incidence and Risk Factors of Calcium Channel Blocker–Related Edema in Hypertensive Patients: A Multicenter Retrospective Cohort StudyPLOS ONE

Dear Dr. Solela,

Thank you for submitting your manuscript to PLOS ONE. After careful consideration, we feel that it has merit but does not fully meet PLOS ONE’s publication criteria as it currently stands. Therefore, we invite you to submit a revised version of the manuscript that addresses the points raised during the review process.

Please respond to the 2 points made by myself and the multiple points made by the 3 reviewers below and including the attachments.  

We look forward to receiving your revised manuscript.

Kind regards,

James M Wright

Academic Editor

PLOS ONE

Journal Requirements:

2. Please amend the manuscript submission data (via Edit Submission) to include author Mulualem Alemayehu Gebreselassie.

3. Please amend your authorship list in your manuscript file to include author Mulualem Alemayehu Gebreselassiea.

[The authors have declared that no competing interests exist.].

We note that one or more of the authors are employed by commercial companies: Yehuleshet Specialty Clinic and ICMC General Hospital.

Additional Editor Comments:

1. Please correct the definition of hypertension to the more accepted 140/90 mmHg realizing that this is an arbitrary number from guidelines.

2. An important statistic that is missing is the number os people who stopped the drug due to the edema.

Reviewers' comments:

Reviewer's Responses to Questions

**Comments to the Author**

1. Is the manuscript technically sound, and do the data support the conclusions?

Reviewer #1: Partly

Reviewer #2: Yes

Reviewer #3: Yes

2. Has the statistical analysis been performed appropriately and rigorously? 

Reviewer #1: I Don't Know

Reviewer #2: Yes

Reviewer #3: Yes

3. Have the authors made all data underlying the findings in their manuscript fully available?

Reviewer #1: Yes

Reviewer #2: Yes

Reviewer #3: Yes

4. Is the manuscript presented in an intelligible fashion and written in standard English?

Reviewer #1: Yes

Reviewer #2: Yes

Reviewer #3: Yes

5. Review Comments to the Author

Reviewer #1: Please see my Comments on the PDF manuscript. I think it's worth publishing, consistent with the mission of PLoS One. I think you need someone else to comment on whether the statistical approach is appropriate (or not). I do suggest reporting adjusted ORs (after defining what adjustment was done, or what "AOR" means), rather than arbitrarily saying one AOR is "significant" but another is not.

I think the main value of the manuscript is to remind readers that edema is frequent and can occur early during CCB treatment, especially with dihydropyridines. It would be even more interesting to know the following, and if this MS is not accepted, perhaps the authors would undertake a prospective study in the same environment to answer these questions:

1. How early can edema occur from CCBs - if one checks before drug initiation (Day 0) could edema be present by Day 1 or Day 2, or only by Day 7 (1 week)?

2. Does edema relate in any way to antihypertensive effect? For example, if there is no change in BP, can edema still occur (assuming patient is taking the drug)???

This raises the question of how authors ascertained whether patients were taking the prescribed drugs, and also of how they measured BP. I have commented about the latter on the MS, but didn't think about adherance until writing this comment.

Also, I have always figured that if I had diabetic or other peripheral neuropathy, edema would be BAD - more chance of failing to notice a cut or abrasion, more "tissue fluid" in which bacteria or fungi could breed, etc. If the authors know anything about this, or could review the issue as part of discussion, I think that could be very interesting and useful to readers - and a real advance in discussion of the issue.

I recommend acceptance, but asking the authors to address these and any other comments. The English is fine, but I recommended consistent active voice over passive voice. That is a trivial issue of style, about which PLoS One may or may not have its own views.

The authors are welcome to my comments on the PDF, and my name and qualifications.

Thomas L. Perry MD, FRCPC (general internal medicine)

Editor, Therapeutics Letter, www.ti.ubc.ca

Clinical Professor

Dept. of Anaesthesiology, Pharmacology & Therapeutics, University of BC, Canada

Reviewer #2: This is an automated report for PONE-D-25-58381. This report was solicited by the PLOS One editorial team and provided by ScreenIT.

ScreenIT is an independent group of scientists developing automated tools that analyze academic papers. A set of automated tools screened your submitted manuscript and provided the report below. Each tool was created by your academic colleagues with the goal of helping authors. The tools look for factors that are important for transparency, rigor and reproducibility, and we hope that the report might help you to improve reporting in your manuscript. Within the report you will find links to more information about the items that the tools check. These links include helpful papers, websites, or videos that explain why the item is important. While our screening tools aim to improve and maintain quality standards they may, on occasion, miss nuances specific to your study type or flag something incorrectly. Each tool has limitations that are described on the ScreenIT website. The tools screen the main file for the paper; they are not able to screen supplements stored in separate files. Please note that the Academic Editor had access to these comments while making a decision on your manuscript. The Academic Editor may ask that issues flagged in this report be addressed. If you would like to provide feedback on the ScreenIT tool, please email the team at ScreenIt@bih-charite.de. If you have questions or concerns about the review process, please contact the PLOS One office at plosone@plos.org.

Reviewer #3: 1. The study presents the results of original research.

Yes, results were specifically obtained following a protocol that was submitted to, and agreed by, a local ethical committee.

2. Results reported have not been published elsewhere.

No, as far as I can ascertain… Another publication is unlikely due to the delay.

3. Experiments, statistics, and other analyses are performed to a high technical standard and are described in sufficient detail.

Almost: The sample size is justified and has been reached, but its justification lacks some details. Indeed, there is an appropriate power (80%), but we do not know what is the hypothesis behind. Is this the precision of the confidence interval for the estimate of the incidence of edema? Or is this the possibility to identify risk factors, i.e. to contrast the incidence or edema between two groups presenting or not the risk factor?

4. Conclusions are presented in an appropriate fashion and are supported by the data.

In general yes, but I would suggest to the authors some changes.

4.1. Definition of hypertension

Such a definition is arbitrary. The risk of cardiovascular events rises continuously for all levels of blood pressure1, and the lowest level could be adopted as well as more conservative ones. The issue behind such choice is obviously the association of a definite hypertension with an indication for pharmacological treatment. As a consequence, the lowest level of the definition will be associated with the largest volume of prescription. Guidelines have progressively moved towards lower levels along decades, from 160 mmHg initially re. systolic blood pressure to 130 mmHg as adopted by the authors. It is well established that guidelines are produced by experts who are full of conflicts of interest with pharmaceutical industry. The lower levels of hypertension definition along time is a likely explanation for the increase of the prevalence of hypertension mentioned by the authors.

Another possibility would be to chose as a definition of hypertension the level of blood pressure that is associated with a certainty that pharmacological treatment is beneficial. Guidelines acknowledge that there is weaker evidence for a benefit in mild hypertension, i.e. between 140 and 160 mmHg of systolic blood pressure. We established that there is no evidence for a favourable risk to benefit ratio in mild hypertension2. As a consequence, it is highly likely that the studied population is overtreated, confronted to side effects of pharmacological treatment without any solid hope for a benefit. This fact must be acknowledged in the introduction and in the discussion.

4.2. Methods of an observational retrospective study.

The strengths of the study are notable. The design is defined around the main objective of estimating the incidence of edema. They use medical files of known quality from two big hospitals with appropriate medical staff, and a supervision of the data extraction. The sampling procedure seems appropriate. Some aspects of the results suggest that there is a strong reality behind them: the relationship between dosage and the rate of edema; the increase of this rate along time of exposure.

The specific limitations of the experimental design must be better acknowledged by the authors.

First, there is no control group to compare the incidence of edema, which could be attributable to the exposure to CCBs or to other factors as well. The authors could have merely chosen to complete their study in comparing edema incidence with hypertensive people never treated by CCBs. The higher rate of edema they observe could be related to the fact that they attribute to CCBs edema that would have occurred under placebo. Placebo control in randomized trial is the best way to attribute causally an event to the treatment exposure, which is not possible here. In a meta-analysis of RCTs on edema with CCBs, authors found that 29% of edema observed on CCBs were also observed on placebo3.

Second, retrospective analyses are exposed to specific biases that must be discussed.

4.3. The coherence of these results with the existing literature is convincing, e.g. the standing position as a risk factor of edema, or the protective effect of ACE inhibitors against edema occurrence, well illustrated elsewhere4.

4.4. The conclusion that physicians should avoid unnecessary interventions is a truism, and has nothing specific to this population, or this treatment. CCBs are among the best BP lowering drugs to reduce the risk of stroke5. So the risk of oedema has to be balanced against the benefit from this drugs relative of stroke. And the majority of patients are likely to have mild hypertension without any solid hope for benefit.

5. The article is presented in an intelligible fashion and is written in standard English.

Yes, at least for a French reader !!!

6. The research meets all applicable standards for the ethics of experimentation and research integrity.

Yes, with informed consent collected for retrospective use of data collected within medical files.

7. The article adheres to appropriate reporting guidelines and community standards for data availability.

Yes. Data are said to be fully available without restriction.

1. MacMahon S, Peto R, Cutler J, et al. Blood pressure, stroke, and coronary heart disease. Part 1, Prolonged differences in blood pressure: prospective observational studies corrected for the regression dilution bias. Lancet 1990; 335(8692): 765-74.

2. Diao D, Wright JM, Cundiff DK, Gueyffier F. Pharmacotherapy for mild hypertension. Cochrane Database Syst Rev 2012; 2012(8): Cd006742.

3. Makani H, Bangalore S, Romero J, et al. Peripheral edema associated with calcium channel blockers: incidence and withdrawal rate--a meta-analysis of randomized trials. J Hypertens 2011; 29(7): 1270-80.

4. Liang L, Kung JY, Mitchelmore B, Cave A, Banh HL. Comparative peripheral edema for dihydropyridines calcium channel blockers treatment: A systematic review and network meta-analysis. J Clin Hypertens (Greenwich) 2022; 24(5): 536-54.

5. Turnbull F. Effects of different blood-pressure-lowering regimens on major cardiovascular events: results of prospectively-designed overviews of randomised trials. Lancet 2003; 362(9395): 1527-35.

6. PLOS authors have the option to publish the peer review history of their article (what does this mean? ). If published, this will include your full peer review and any attached files.

**Do you want your identity to be public for this peer review?** For information about this choice, including consent withdrawal, please see our Privacy Policy .

Reviewer #1: **Yes:** Thomas L. Perry MD, FRCPC

Reviewer #2: No

Reviewer #3: **Yes:** François GUEYFFIER

---

## [Author Response · Author response to Decision Letter 1]

4 Mar 2026

Academic Editor Comments

Authors’ Responses: Thank you for your comment. We have revised the manuscript and all associated files to fully comply with PLOS ONE’s style and file-naming requirements.

Please amend the manuscript submission data (via Edit Submission) to include author Mulualem Alemayehu Gebreselassie.

Authors’ Responses: Thank you for the notification. We have amended the manuscript submission data (via Edit Submission) to include author Mulualem Alemayehu Gebreselassie.

Please amend your authorship list in your manuscript file to include author Mulualem Alemayehu Gebreselassie.

Authors’ Responses: Thank you for pointing that out. We have updated the authors' list at the beginning of the manuscript and revised the section on authors' contributions as well.

Please provide an amended Funding Statement declaring this commercial affiliation, as well as a statement regarding the Role of Funders in your study.

Authors’ Responses: Thank you for this request. We would like to note that the authors with commercial affiliations made purely scholastic contributions. The manuscript has been revised accordingly.

Please also provide an updated Competing Interests Statement declaring this commercial affiliation, along with any other relevant declarations relating to employment, consultancy, patents, products in development, or marketed products, etc.

Authors’ Responses: Thank you for this request. We would like to note that there is no competing interest to be mentioned regarding the commercial companies mentioned. The manuscript has been revised accordingly.

Your ethics statement should only appear in the Methods section of your manuscript.

Authors’ Responses: Thank you for the suggestion. We have revised the manuscript so that the Ethical Statement now appears at the beginning of the Materials and Methods section.

Please include captions for your Supporting Information files at the end of your manuscript, and update any in-text citations to match accordingly.

Authors’ Responses: Thank you for this helpful comment. Captions for all Supporting Information files have been added at the end of the manuscript, and all in-text citations have been revised accordingly to ensure consistency.

Please correct the definition of hypertension to the more accepted 140/90 mmHg, realizing that this is an arbitrary number from guidelines.

Authors’ Responses: Thank you for your comment. We have updated the definition of hypertension to the widely accepted threshold of 140/90 mmHg, acknowledging that this value is guideline-based and somewhat arbitrary.

An important statistic that is missing is the number of people who stopped the drug due to the edema

Authors’ Responses: Thank you for your comment. That was an important part of the study at its start. However, since the study required a diagnosis of CCB-related edema to be made after physician evaluation, it was challenging to include these patients solely on their reports. To the investigators' best knowledge, the number of patients who reported discontinuation before the visit was low, as follow-up visits are monthly.

Reviewer 1 Comments

I do suggest reporting adjusted ORs (after defining what adjustment was done, or what "AOR" means), rather than arbitrarily saying one AOR is "significant" but another is not.

Authors’ Responses: Thank you for your valuable feedback. We have now defined AOR in the “Data Analysis” section as the odds ratio from multivariable logistic regression adjusted for potential confounders, ensuring all associations are interpreted based on proper adjustment rather than arbitrary significance.

How early can edema occur from CCBs - if one checks before drug initiation (Day 0) could edema be present by Day 1 or Day 2, or only by Day 7 (1 week)?

Thank you for your comment. No patients developed edema within the first week of CCB therapy. None of the patients experienced edema during the first week of CCB therapy. Overall, 38.7% developed peripheral edema, occurring at a mean of 10.5 ± 8.3 weeks after treatment initiation. We have highlighted these changes in the revised manuscript.

Does edema relate in any way to antihypertensive effect? For example, if there is no change in BP, can edema still occur (assuming patient is taking the drug)???

Authors’ Responses: Thank you for this insightful comment. We have analyzed the relevant data accordingly and have incorporated these results into the revised manuscript to clarify this distinction.

Reviewer 2 Comments

We did not find a study flow chart of excluded observations. We strongly recommend using flow charts because they provide an overview of the study design and more information about attrition. If you included a study flowchart in your supplemental files, we apologize for missing it. Our tool is not able to screen separate supplemental files.

Authors’ Responses: Thank you for your comment. We acknowledge the importance of a study flowchart to summarize participant inclusion and attrition. We have now provided a study flowchart in the revised manuscript.

Sentence about attrition: not detected. Please provide information about the drop-out of subjects, or loss of animals or samples. This could be done using a study flow chart, or described in the text.

Authors’ Responses: Thank you for your comment. This is not applicable to our study, as it was a retrospective cohort study and all eligible patient records were available for analysis. Therefore, no drop-outs, loss of subjects, or missing data occurred. However, we have provided a study flowchart to illustrate the selection of participants and the study population.

Blinding not detected. Please specify whether blinding was used at various phases of the experiment (e.g., blinding of patients, caregivers, outcome assessments, data analysis).

Authors’ Responses: Thank for your comment. Blinding was not applicable in our study.

An explicit section about the limitations of this study was not found. We encourage authors to include a paragraph in the discussion that addresses study limitations. Every study has limitations. Describing these limitations helps readers to understand and contextualize the research.

Authors’ Responses: We appreciate your suggestion. A dedicated paragraph has been added at the end of the Discussion section to transparently address the limitations of our study.

Reviewer 3 Comments

Is this the precision of the confidence interval for the estimate of the incidence of edema? Or is this the possibility to identify risk factors, i.e. to contrast the incidence or edema between two groups presenting or not the risk factor?

Authors’ Responses: Thank you for your comment. While we did not report a confidence interval for the overall incidence of edema, the 95% confidence intervals were used to assess the precision of the associations between potential risk factors and edema.

Conclusions: I would suggest to the authors some changes.

Authors’ Responses: Thank you for your feedback. We have made some adjustments to the conclusion.

The definition of hypertension is arbitrary.

Authors’ Responses: Thank you for your feedback. We have updated the definition of hypertension to the widely accepted threshold of 140/90 mmHg, acknowledging that this value is guideline-based and somewhat arbitrary.

The specific limitations of the experimental design must be better acknowledged by the authors.

- First, there is no control group to compare the incidence of edema

- Retrospective analyses are exposed to specific biases that must be discussed.

Authors’ Responses: Thank you for your comment. We greatly appreciate this input and have now included a dedicated paragraph on the limitations of our study in the Discussion section of the revised manuscript.

The conclusion that “physicians should avoid unnecessary interventions” is a truism, and has nothing specific to this population, or this treatment.

CCBs are among the best BP lowering drugs to reduce the risk of stroke. So the risk of oedema has to be balanced against the benefit from these drugs relative of stroke. And the majority of patients are likely to have mild hypertension without any solid hope for benefit.

Authors’ Responses: We found your comment to be valid, as the statement was somewhat vague. Accordingly, we have revised the wording in the Discussion section.

---

## [Editor Report · Decision Letter 1]

9 Mar 2026

Incidence and Risk Factors of Calcium Channel Blocker–Related Edema in Hypertensive Patients: A Multicenter Retrospective Cohort Study

PONE-D-25-58381R1

Dear Dr. Solela,

We’re pleased to inform you that your manuscript has been judged scientifically suitable for publication and will be formally accepted for publication once it meets all outstanding technical requirements.

Kind regards,

James M Wright

Academic Editor

PLOS One
---

## [Editor Report · Acceptance letter]

PONE-D-25-58381R1

PLOS One

Dear Dr. Solela,

I'm pleased to inform you that your manuscript has been deemed suitable for publication in PLOS One. Congratulations! Your manuscript is now being handed over to our production team.

Kind regards,

on behalf of

Professor James M Wright

Academic Editor

PLOS One